# Resonant Transmission Line Method for Unconventional Fibers

**Anthony C. Boucouvalas [1,\*], Christos D. Papageorgiou [2], Eurypides Georgantzos [1] and Theophanes E. Raptis [1,3,4]**

[1] Department of Informatics and Telecommunications, University of Peloponnese, 22131 Tripoli, Greece; e.georgantzos@gmail.com (E.G.); rtheo@dat.demokritos.gr (T.E.R.)

[2] Department of Electrical Engineering, National Technical University of Athens, 15780 Athens, Greece; chrpapa@hol.gr

[3] National Center for Science and Research "Demokritos", Division of Applied Technologies, 15341 Athens, Greece

[4] Physical Chemistry Lab., Chemistry Department, National Kapodistrian University of Athens, 15784 Athens, Greece

\* Correspondence: acb@uop.gr

**Abstract:** We provide a very general review of the resonant transmission line method for optical fiber problems. The method has been found to work seamlessly for a variety of difficult problems including elliptical and eccentric core fibers as well as "holey" photonic crystal fibers. This new version has been shown to offer great versatility with respect to cases of unconventional, inhomogeneous index profiles.

**Keywords:** transverse resonance; optical fibers; mode propagation constant; electromagnetic waves; elliptical core fibers; eccentric core fibers; photonic crystal fibers; refractive index

## 1. Introduction

The method of resonant transmission lines (RTL) has evolved from some original observations around the generic theme of transverse resonance in direct association with the standard theory of the solutions of the general telegrapher's equation in transmission lines. The particular advantage of the method is that it offers the possibility of reducing even full three-dimensional problems into one its dimensional equivalent, at least wherever a full variable separation is possible due to the particular symmetries of a given coordinate system. A full historical review of the use of transmission lines for modeling electromagnetic problems would require too much space but we can provide some important milestones in chronological order in brief.

The transmission line theme can be traced as far back as in the old treatise by Schelkunoff [1]. Further physical justification is offered by a seminal paper by Marcuvitz and Schwinger [2] where we find a first version of the transverse resonance condition as a means to satisfy certain boundary conditions via appropriate impedance matching. Extensive use of the telegrapher's equation is also found in Ramachandran [3], Gallawa [4], Clarricoats and Oliner [5,6], Yoneyama [7], Borneman and Arndt [8], Tao [9], Shigesawa and Tsuji [10] as well as Dahl et al. [11]. More recently, Moshonas et al. [12] also used transmission line methods for the study of photovoltaic cells. An important comparison between transverse stationary modes and their similarity with quantum wells is mentioned by Bialynicki-Birula [13] and has led to the application of the RTL method by some of the authors in Schrödinger type and general Sturm–Liouville problems [14–16]. An early application of similar ideas in optical fibers can be traced in a paper by Yeh and Lindgren [17]. Tamir also discussed guided waves based on impedance matching in Reference [18], while later Carlin and Zmuda [19] applied

transfer matrix methods for inhomogeneous fibers. More recently, Mencarelli and Rossi [20] used transverse resonance for studying multilayered photonic crystals.

The modern RTL method was originally introduced by Papageorgiou and Boucouvalas [21–26] in simple cases of standard step index and radial index profiles. Lately, it was also applied successfully in the case of superlattices with periodic potentials [27] which are similar to a certain class of photonic crystal fibers (PCFs) or Bragg fibers in general. Several other cases of unusual refractive index profiles have been successfully treated with the same method [28–31]. In the present work, we expose more recent developments in an effort to expand the validity of the RTL method in all unconventional fiber models.

We shall heretofore refer to all the cylindrical optical fibers as the class of conventional optical fibers (COFs) for as long as they can be separated in a set of $n$ very thin successive cylindrical layers of average radius $r$ with uniform refractive index values given as $n_i$. Indices of successive layers will be allowed in general to be different for each step varying as $n_i = n(r_i)$. The resulting, total index profile $n(r)$ variation from the center of the fiber up to the limit outer air medium, completely defines the propagation properties inside the fiber.

We shall also separate another class of unconventional optical fibers (UOFs) as the ones in which at least in some of its successive thin cylindrical layers present an additional variation of the total index profile along its radial coordinate $\varphi$ in the form $\eta(r,\varphi)$. Such cases include elliptic core, non-symmetric or eccentric core fibers, and in general, all cases of not strictly circular core fibers. In these cases, any discretization into thin cylindrical layers that "cuts" through the core and cladding, results in a variation of the refractive index along $\varphi$. In Figure 1, we show such a case of a thin, circular cylindrical layer cutting an elliptic core fiber. As is evident in the schematic, the variation is analogous to the arc length of the circular sector cut by the ellipse for each discretization step.

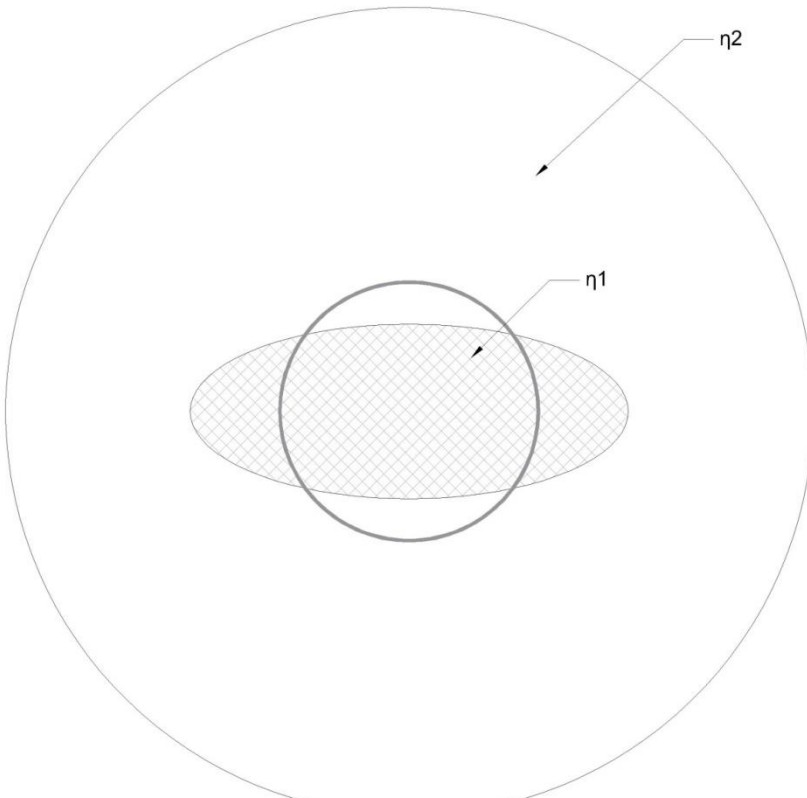

**Figure 1.** Example of the alternating character of the local index value from a thin shell radial discretization.

Additionally, photonic fibers made of silica with a set of small air holes around their centers will have many cylindrical layers of varying refractive indexes along $\theta$. In Figure 2, we show an example of a PCF with a hexagonal lattice of 6 rows of air holes.

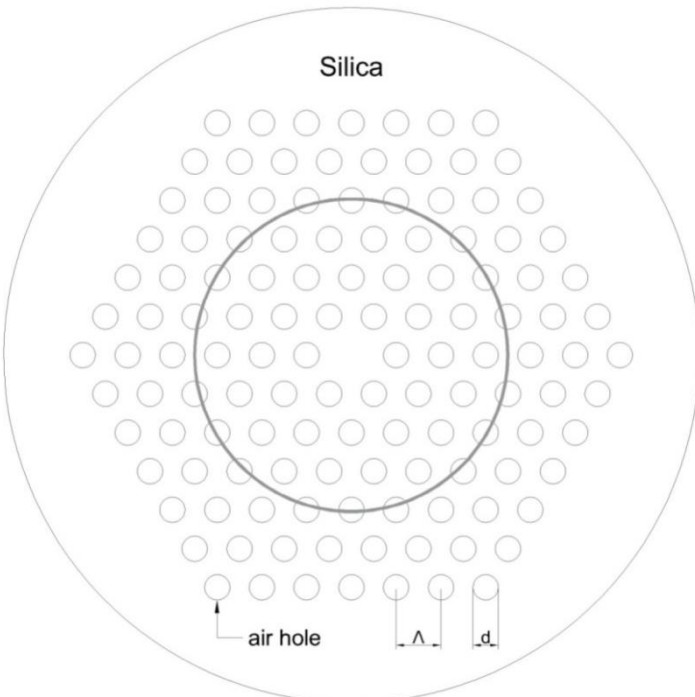

**Figure 2.** Schematic depiction of the alternating step index resulting from radial discretization in a standard photonic crystal fiber (PCF).

Again, radial discretization with thin, circular cylindrical layers results in alternating cuts through air holes and silica of a PCF as depicted in the schematic. The main aim in the present work is to develop the original RTL method so as to be able to include all such UOF cases via an appropriate transformation to mathematically equivalent COF cases.

In the following sections, we first introduce the exact mathematical formulation of the discretization process for Maxwell's equations in the context of the RTL method in the following section. We also show how to perform an algebraic decoupling of the resulting transmission line equations and the transfer function method for computing the actual fields from the eigenvalues obtained. In Section 3, the same tools are applied in four more difficult cases of UOF with either, asymmetry or eccentricity of the central core. We also perform the same analysis for elliptic and rectangular cores as well as for the PCF case.

## 2. Mathematical Equivalence of Homogeneous Circular Cylindrical Layers to Electric Transmission Lines

The basis for the application of the previously introduced RTL method is the radial discretization of all cylindrical fibres via a separation into a succession of thin cylindrical layers, each one with its own constant refractive index $n$. These layers can be made to extend outside of the cladding in order to take into consideration the effect of the surrounding air ($n = 1$). Each thin cylindrical layer could have thickness $\delta r$ proportional to each average radius $r$ which means that given discrete steps as $\delta r = r_2 - r_1$ with $r = \frac{r_1 + r_2}{2}$ one has

$$\frac{r_2 - r_1}{r_2 + r_1} = \frac{c}{2} => \begin{cases} \frac{1 + \frac{c}{2}}{1 - \frac{c}{2}} r_1 = r_2 \ (out) \\ \frac{1 - \frac{c}{2}}{1 + \frac{c}{2}} r_2 = r_1 \ (in) \end{cases} \tag{1}$$

For any such circular cylindrical layer Maxwell equations (for a constant wavelength, i.e., constant frequency "$\omega$") can be written in their standard form as

$$\begin{cases} \nabla X \vec{E} = -j\omega\mu_0\vec{H} \\ \nabla X \vec{H} = j\omega\varepsilon_0 n^2(r)\vec{E} \end{cases} \tag{2}$$

Taking into consideration the relations $\omega\mu_0 = k_0 z_0$ and $\omega\varepsilon_0 = \frac{k_0}{z_0}$ where $k_0 = \frac{\omega}{c}$, $z_0 = 120\pi$ and replacing $z_0\vec{H}$ with $\vec{H}$ in order $\vec{E}$ and $\vec{H}$ to have the same units $(V/m)$, Maxwell equations become then

$$\begin{cases} \nabla X \vec{E} = -jk_0\vec{H} \\ \nabla X \vec{H} = jk_0 n^2(r)\vec{E} \end{cases} \tag{3}$$

In circular cylindrical geometry of coordinates $(r, \varphi, z)$ the following set of three partial differential equations can be derived by the first vector Maxwell equation as

$$\begin{cases} \frac{1}{r}\frac{\partial E_z}{\partial \varphi} - \frac{\partial E_\varphi}{\partial z} = -jk_0 H_r \\ \frac{\partial E_r}{\partial z} - \frac{\partial E_z}{\partial r} = -jk_0 H_\varphi \\ \frac{1}{r}\frac{\partial(rE_\varphi)}{\partial r} - \frac{1}{r}\frac{\partial E_r}{\partial \varphi} = -jk_0 H_z \end{cases} \tag{4}$$

Applying a Fourier Transform along "z" and "$\varphi$" with wave numbers "$\beta$" and "$l$", where $l$ is integer (because along "$\varphi$" we have Fourier series of period $2\pi$), the set (4) becomes:

$$\begin{cases} \frac{jl}{r}\overline{E_z} - j\beta\overline{E_\varphi} = -jk_0\overline{H_r} \\ j\beta\overline{E_r} - \frac{\partial\overline{E_z}}{\partial r} = -jk_0\overline{H_\varphi} \\ \frac{1}{r}\frac{\partial(r\overline{E_\varphi})}{\partial r} - \frac{jl}{r}\overline{E_r} = -jk_0\overline{H_z} \end{cases} \tag{5}$$

In (5) we use new variables $\overline{E_r}$, $\overline{E_\varphi}$, $\overline{E_z}$, $\overline{H_r}$, $\overline{H_\varphi}$, $\overline{H_z}$ to denote the Fourier Transforms of the respective electromagnetic field components. Furthermore, replacing $\beta$ and $r$ by their reduced variables according to the following relations:

$$\begin{cases} \frac{\beta}{k_0} => \beta \\ rk_0 => r \end{cases}$$

Then (5) takes the form

$$\begin{cases} \frac{jl}{r}\overline{E_z} - j\beta\overline{E_\varphi} = -j\overline{H_r} \\ j\beta\overline{E_r} - \frac{\partial\overline{E_z}}{\partial r} = -j\overline{H_\varphi} \\ \frac{1}{r}\frac{\partial(r\overline{E_\varphi})}{\partial r} - \frac{jl}{r}\overline{E_r} = -j\overline{H_z} \end{cases} \tag{6}$$

Following a similar approach, the second Maxwell vector Equation (3) can be written in the form

$$\begin{cases} \frac{jl}{r}\overline{H_z} - j\beta\overline{H_\varphi} = jn^2(r)\overline{E_r} \\ j\beta\overline{H_r} - \frac{\partial\overline{H_z}}{\partial r} = jn^2(r)\overline{E_\varphi} \\ \frac{1}{r}\frac{\partial(r\overline{H_\varphi})}{\partial r} - \frac{jl}{r}\overline{H_r} = jn^2(r)\overline{E_z} \end{cases} \tag{7}$$

Furthermore, following a cumbersome analysis, it is possible to prove that the system of Equations (6) and (7) can be transformed in a set of four differential Equation (8), relating the equivalent "voltage' and "current" functions $V_M$, $I_M$, $V_E$, $I_E$ defined as follows:

$$V_M = \frac{l\overline{H_\varphi} + \beta r \overline{H_z}}{jF}$$

$$I_M = \frac{r\overline{H_r}}{j} = \frac{\beta r \overline{\varphi} - l\overline{z}}{j}$$

$$V_E = \frac{l\overline{\varphi} + \beta r \overline{E_z}}{F}$$

$$I_E = n^2 r \overline{E_r} = l\overline{H_z} - \beta r \overline{H_\varphi}$$

where we use the notation $F = \frac{(\beta r)^2 + l^2}{r}$

$$\begin{cases} \frac{\partial V_M}{\partial r} = -\frac{\gamma^2}{jF} I_M - jMI_E \\ \frac{\partial I_M}{\partial r} = -jFV_M \\ \frac{\partial V_E}{\partial r} = -\frac{\gamma^2}{jn^2F} I_E - jMI_M \\ \frac{\partial I_E}{\partial r} = -jn^2 FV_E \end{cases} \tag{8}$$

In (8) we introduced the total propagation factor $\gamma^2 = \frac{l^2}{r^2} + \beta^2 - n^2$ and the auxiliary function $M = \frac{2l\beta}{[(\beta r)^2 + l^2]F}$.

At this point it is noticed that $V_M$, $I_M$, $V_E$, $I_E$ are continuous functions at the boundaries because the tangential components of electric and magnetic fields $\vec{H_\varphi}$ $\vec{H_z}$ and $\vec{E_\varphi}$ $\vec{E_z}$ on the cylindrical surface are continuous functions passing the boundaries' of the cylindrical layer. Using the previous relations, the Fourier Transforms of the electromagnetic field components along ($r$, $l$, $\beta$) can be expressed as functions of their equivalent "voltages" and "currents" functions with the auxiliary relations

$$\overline{H_r} = \frac{jI_M}{r}, \; \overline{E_r} = \frac{I_E}{n^2 r}$$

$$\overline{H_\varphi} = jlV_M / r - \frac{\beta}{F} I_E$$

$$\overline{E_\varphi} = lV_E / r + j\frac{\beta}{F} I_M$$

$$\overline{H_z} = \frac{l}{Fr} I_E + j\beta V_M$$

$$\overline{z} = -j\frac{l}{Fr} I_M + \beta V_E$$

It becomes evident by inspection that the final Equation (8) represents two coupled electric transmission lines.

## 2.1. Decoupling the Transmission Line Equations

The prescribed set of Equation (8) constitutes a homogeneous set of ordinary differential equations of $r$ and considering that the all vectors $[V_M, I_M, V_E, I_E]$ can be turned into exponential functions of $r$ given by $V_M = V_M e^{\xi r}$, $I_M = I_M e^{\xi r}$, $V_E = V_E e^{\xi r}$, $I_E = I_E e^{\xi r}$, where $V_M$, $I_M$, $V_E$, $I_E$ are constants, i.e., not functions of $r$. Thus, the system (8) can be transformed in an algebraic set of the following four equations

$$\begin{cases} \xi V_M = -\frac{\gamma^2}{jF} I_M - jMI_E \\ \xi I_M = -jFV_M \\ \xi V_E = -\frac{\gamma^2}{jn^2F} I_E - jMI_M \\ \xi I_E = -jn^2 FV_E \end{cases} \tag{9}$$

Replacing $I_M = -\frac{jF}{\xi} V_M$, $I_E = -\frac{jn^2 F}{\xi} V_E$, we obtain a set of two homogeneous equations

$$\begin{cases} \xi V_M = \frac{\gamma^2}{\xi} V_M - M\frac{n^2 F}{\xi} V_E \\ \xi V_E = \frac{\gamma^2}{n^2 \xi} V_E - M\frac{F}{\xi} V_M \end{cases} \quad \text{or} \quad \begin{cases} \xi^2 V_M = \gamma^2 V_M - n^2 MFV_E \\ \xi^2 V_E = \gamma^2 V_E - MF \end{cases}$$

This then leads to the eigenvalue equations

$$\begin{cases} (\xi^2 - \gamma^2)V_M + n^2 MFV_E = 0 \\ MFV_M + (\xi^2 - \gamma^2)V = 0 \end{cases}$$

From the standard form of the eigenvalue problem we obtain through the determinant differential equations as follows

$$\left(\xi^2 - \gamma^2\right)^2 - n^2 M^2 F^2 = 0, \text{ or } \xi^2 = \gamma^2 \pm nMF \tag{10}$$

Hence the system has two eigenvalues and two mutually excluded or "normal" eigenvectors. The eigenvectors will be found by replacing $\xi^2$ by its value. Thus, for $\xi^2 = \gamma^2 - nMF$, $n^2 MFV_E - nMFV_M = 0 => V_M = nV_E$ and the eigenvector is $V_S = V_M + nV_E$ While for $\xi^2 = \gamma^2 + nMF$, $V_M = -nV_E$ and the eigenvector becomes $V_d = V_M - nV_E$. Their respective "current" eigenvectors are related as $\frac{I_M}{I_E} = \frac{V_M}{n^2 V_E} = \frac{1}{n}$, thus $I_M = \frac{I_E}{n}$ and $I_s = I_M + \frac{I_E}{n}$, $I_d = I_M - \frac{I_E}{n}$. Since the auxiliary $M$ function has the sign of $l$, the set $(V_s, I_s)$, for $l = -l$ becomes equal to the set $(V_d, I_d)$. Thus, we can consider as a unique solution for the set $(V_s, I_s)$ and the integer '$l$ 'varies $-\infty \text{ } to + \infty$, and of course $\xi^2 = \gamma^2 - nMF$:

$$\begin{cases} \frac{\partial V_s}{\partial r} = -\frac{\xi^2}{jF}I_s \\ \frac{\partial I_s}{\partial r} = -jFI_s \end{cases} \tag{11}$$

Furthermore, $V_s$, $I_s$ should be continuous functions at their boundaries although $n(r)$ varies from layer to layer. This is achieved via the adjustment $V_s = V_M + nV_E = 2V_M$ and $I_s = I_M + \frac{I_E}{n} = 2I_M$, which are continuous functions of $r$ by definition.

$$\begin{cases} \frac{\partial V_M}{\partial r} = -\frac{\xi^2}{jF}I_M \\ \frac{\partial I_M}{\partial r} = -jFI_M \end{cases} \tag{12}$$

Another option for achieving continuity is to consider the functions $V_{ss} = \frac{V_M}{n} + V_E$ and $I_{ss} = nI_M + I_E$. In this case, $V_{ss} = 2V_E$ and $I_{ss} = 2I_E$ are also continuous. Thus

$$\begin{cases} \frac{\partial V_E}{\partial r} = -\frac{\xi^2}{jn^2 F}I_E \\ \frac{\partial I_E}{\partial r} = -jFn^2 I_E \end{cases} \tag{13}$$

Thus, the set of two coupled transmission lines (9) is equivalent to two independent transmission lines (12) and (13).

The two waves represented by the equations of transmission lines (12) and (13), are geometrically normal because the first is related to the magnetic field and the second to the electric field that are geometrically normal for transmitted EM waves. This property is an inherent property of EM modes in optical fibers related to birefringence phenomena. However, the $\beta$ respective values, for any mode, are always found to be very close and can be considered as practically equal.

## 2.2. Equivalent Circuits for Cylindrical Layers, Boundary Conditions, and Birefringence

Taking into consideration the transmission line theory, it can be proved that each layer of infinitesimal thickness $\delta r$ is equivalent to a T-circuit as the one shown in Figure 3

$$\begin{cases} Z_B = \frac{\xi}{jF}\tanh\left[\frac{(\xi\delta r)}{2}\right] \\ Z_p = \frac{\xi}{jF\sinh(\xi\delta r)} \end{cases}$$

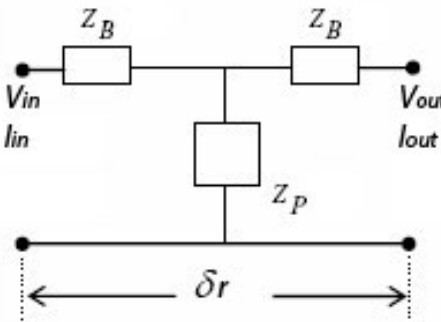

**Figure 3.** The equivalent quadrupole for each cylindrical sector.

For $\xi\delta r \ll 1$ the impedances can be approximated by the equivalent relations

$$
\begin{cases}
Z_B = \dfrac{\xi^2\left(\frac{\delta r}{2}\right)}{jF} \\[2mm]
Z_p = \dfrac{1}{jF\delta r}
\end{cases}
\tag{14}
$$

If $\xi^2 > 0$, both $Z_B$, $Z_p$ are "capacitive" reactances, for $\xi^2 < 0$ however $Z_B$ becomes "inductive" reactance. For $(V_E, I_E)$ the approximate respective impedances of the T-circuit are given as

$$
\begin{cases}
Z'_B = \dfrac{\xi^2\left(\frac{\delta r}{2}\right)}{jn^2F} \\[2mm]
Z'_p = \dfrac{1}{jn^2F\delta r}
\end{cases}
\tag{15}
$$

As previously stated, the functions $(V_M, I_M)$ of each layer are continuous at the cylindrical boundaries of the layer, thus if we divide the fiber (including a sufficient number of air layers) in successive thin layers and replace them by their equivalent T-circuits, an overall lossless transmission line is formed with only reactive elements. For given "$l$", the "$\beta$" values that lead to the resonance of the overall transmission line are the eigenvalues of the whole optical fiber.

When a transmission line is in resonance, at any arbitrary point $r_0$ of the line, the sum of reactive impedances arising from the successive T-circuits on the left and right sides of $r_0$ should be equal to zero, thus the equation giving the eigenvalues of the transmission line is the following:

$$
\left\{ \dot{Z}_{L.r_0} + \dot{Z}_{R.r_0} = 0 \right.
\tag{16}
$$

Equation (16) provides the eigenvalues "$\beta$" for a given "$l$", where $\dot{Z}_{L.r_0}$, $\dot{Z}_{R.r_0}$ are the overall reactive impedances of successive T-circuits on the left and right of $r_0$, using Equations (14) or (15). The value of $r_0$ is usually given by the core radius. For the same "$l$" the Equations (14) and (15) give usually slightly different values of '$\beta$'. This phenomenon is called "Birefringence". For circular step index fibers, the birefringence is negligible; however, for elliptic fibers and fibers of any other non-circular cores, the birefringence phenomenon could be not negligible.

In order to calculate the overall reactive impedances on the left and right of $r_0$ we should find the impedances for $r \to 0$ and for $r \to \infty$. As we proceed to 0 or to $\infty$, the remaining piece of transmission line becomes "homogeneous", i.e., its overall reactive impedance is equal to its characteristic impedance given by $Z = \frac{\xi}{jF}\left(or\ \frac{\xi}{jn^2F}\right)$. Then we must have

$$
r \to \infty : F \to \beta^2 r,\ MF \to 0,\ \xi \to \sqrt{\beta^2 - n^2} \to Z_{r \to \infty} = 0
$$
$$
r \to 0 : F \to \frac{l^2}{r},\ \xi \to \frac{l}{r} \to Z_{r \to 0} = \frac{1}{j|l|}\left(or\ \frac{1}{jn^2|l|}\right)
$$

For $l = 0$ $Z_{r \to 0} = \infty$ (open circuit at the center of the equivalent transmission line) It is useful to notice that there is an equivalence between our formulation and the classic formulation modes of optical fibers. In particular, for $l = 0$, the modes $(V_M, I_M)$ are the TM modes, while the modes $(V_E, I_E)$ are the TE modes. For $l > 0$, the modes $(V_M, I_M)$ are the HE modes, while the modes $(V_E, I_E)$ are their HE birefringence modes. For $l < 0$ the modes $(V_M, I_M)$ are the EH modes, while the modes $(V_E, I_E)$ are their EH birefringence modes. For any given $l$, using the resonance technique the $\beta$ values of the two birefringence modes can be calculated. The Equation (16) is given as a MATLAB function in Appendix A.

Let us consider for example a step-index fiber of $n_1 = 1.54$, $n_2 = 1.47$ the $V_M$, $V_E$, fundamental modes for $V = 3.3$, can be calculated and their $\beta/k_0$ values are respectively 1.518934962534846 and 1.518340184686295, hence their birefringence is equal to 0.0004947 or 0.0391%. The $\beta/k_0$ value for the equivalent mode $V_{eq}$ was also calculated and was equal to 1.518638548412019 (that is approximately equal to the mean value of the previous $\beta/k_0$ values), while the $\beta/k_0$ value calculated classically by Bessel functions is equal to 1.518642063686336. These $\beta$ values are very close differing only 0.0002315%.

In the following Figure 4, the normalized birefringence of the step-index fibers for $n_1 = 1.54$, $n_2 = 1.47$, and of $n_1 = 1.475$, $n_2 = 1.47$ as functions of V are shown.

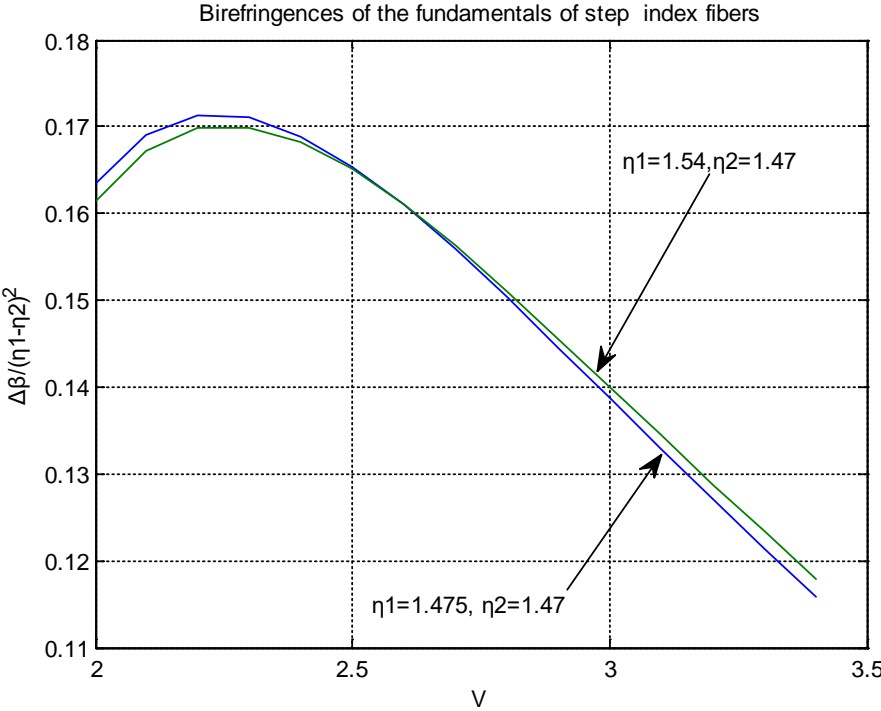

**Figure 4.** Normalized birefringence of two step-index fibers with different refractive indexes as functions of their parameters V.

We notice that for any V, the normalized birefringence is almost proportional to $\Delta n^2 = (n_1 - n_2)^2$, thus the birefringence of step-index fibers of very small $\Delta n$ is negligible. For instance, for a value of V = 2.4, and $\Delta n = 1.54 - 1.47 = 0.07$, the birefringence is found to be $0.168 \times 0.0049 = 0.0008232$ or ~0.055% on the average $\beta$, while for $\Delta n = 1.475 - 1.47 = 0.005$, the birefringence becomes $0.168 \times 0.000025 = 0.000042$ or ~0.0028% on the average $\beta$. What is remarkable is that our method is sensitive and calculates it.

### 2.3. Calculating "Voltages" $V_M$, $V_E$ and "Currents" $I_M$, $I_E$ and Resulting Fields

For any given $l$, using the resonance technique the $\beta$ values of the two birefringence modes can be calculated. These $\beta$ values are practically the same, thus we can consider them as equal or we can consider as the proper value of $\beta$ the mean value of the two modes. Taking $V_M = 1$ at the center point

of the fiber ($r = 0$), the respective value of $I_M$ at the same point can be calculated by the respective terminal impedance. Using the matrix relations between input–output for the equivalent successive T-circuits, the values of $V_M$ and $I_M$ at the rest thin cylindrical layers can be calculated. In fact, from the general theory of the telegrapher's equation we know that the inputs and outputs are associated via a transfer matrix as follows

$$
\begin{aligned}
[V_{out} I_{out}] &= \begin{pmatrix} \cosh(\xi(r) \cdot \delta r) & Z(r) \cdot \sinh(\xi(r) \cdot \delta r) \\ \sinh(\xi(r) \cdot \delta r)/Z(r) & \cosh(\xi(r) \cdot \delta r) \end{pmatrix} \begin{bmatrix} V_{in} \\ I_{in} \end{bmatrix} \\
&\approx \begin{pmatrix} 1 & Z(r) \cdot (\xi(r) \cdot \delta r) \\ (\xi(r) \cdot \delta r)/Z(r) & 1 \end{pmatrix} \begin{bmatrix} V_{in} \\ I_{in} \end{bmatrix} \\
&= \begin{pmatrix} 1 & \xi^2(r) \cdot \delta r/jF(r) \\ jF(r) \cdot \delta r & 1 \end{pmatrix} \begin{bmatrix} V_{in} \\ I_{in} \end{bmatrix}
\end{aligned}
\tag{17}
$$

In Equation (17), the characteristic impedance should be taken as $Z(r) = \xi(r)/jF(r)$ to fit with the previous analysis. Using the relations $nV_E = V_M$ and $nI_M = I_E$ the respective values of their birefringence partners can also be calculated for every thin cylindrical layer $r_i$. Finally, we obtain the actual fields via the relations

$$
\overline{H_r} = \frac{jI_M}{r}, \ \overline{E_r} = \frac{I_E}{n^2 r}
$$
$$
\overline{H_\varphi} = jlV_M / r - \frac{\beta}{F} I_E
$$
$$
\overline{E_\varphi} = lV_E / r + j\frac{\beta}{F} I_M
$$
$$
\overline{H_z} = \frac{l}{Fr} I_E + j\beta V_M
$$
$$
\overline{z} = -j\frac{l}{Fr} I_M + \beta V_E
$$

A very useful field component for optical fibers is the value of the overall electric field at any thin cylindrical layer of average radius $r$ that can be calculated by the formula:

$$
\left| \vec{E}(r) \right|^2 = |\overline{E_r}|^2 + |\overline{E_\varphi}|^2 + |\overline{E_z}|^2
$$

After some algebra, this leads to the formula

$$
|\vec{E}(r)|^2 = \left[ \frac{(\beta r)^2 + l^2}{(nr)^2} \right] |V_M|^2 + \left[ \frac{1}{(\beta r)^2 + l^2} + \frac{1}{(nr)^2} \right] |I_M|^2
$$

In the next section, we extend our analysis in certain UOF cases.

## 3. Unconventional Fibers

The refractive index $n(r,\varphi)$ of the fiber with a UOF profile in general can be described as a function of both $r$ and $\varphi$. Each cylindrical layer of an average radius $r$ is considered to have a local value $\eta(\varphi)$ for $r = \frac{r_1 + r_2}{2}$. Again, we make use of the generic form of Maxwell equations as in Equation (2) of the previous section. Fourier Transforming the first vector equation gives the same set of equations as in (6), since there is no difference with the UOF case, however the second Maxwell vector equation due to the presence of the general $n$ function should now be written as

$$
\begin{cases}
\frac{jl}{r}\overline{H_z} - j\beta\overline{H_\varphi} = jn(l)^2 \otimes \overline{E_r} \\
j\beta\overline{H_r} - \frac{\partial \overline{H_z}}{\partial r} = jn(l)^2 \otimes \overline{E_\varphi} \\
\frac{1}{r}\frac{\partial(r\overline{H_\varphi})}{\partial r} - \frac{jl}{r}\overline{H_r} = jn(l)^2 \otimes \overline{E_z}
\end{cases}
\tag{18}
$$

The symbol $\otimes$ means convolution arising by the product of two functions of the variable $\varphi$. In the following paragraphs it will be shown how to escape this mathematical difficulty for the usual unconventional optical fibers.

### 3.1. UOF with Non-Circular, Non-Symmetric, or Eccentric Cores

For unconventional fibers of non-circular cores there are a set of circular layers where the refractive index varies between the inner and outer core and cladding values, respectively. In any such case, the function $n(\varphi)^2$ is a sum of a steady component $n^2$ and a periodic function of $\varphi$ of period $2\pi$ thus can be written as a Fourier series $n(\varphi)^2 = n^2 + \sum_{-\infty}^{+\infty} N_k \exp(jk\varphi)$. Taking into consideration that the convolution of the product of an exponential function $\exp(jk\varphi)$ with any function $A(\varphi)$ of a Fourier Transform $A(l)$ is equal to $A(l + k)$, i.e., the convolution generates "harmonics". The function $n(\varphi)^2$ is in a set of cylindrical thin layers, a sum of step functions alternating between the values $n_1^2$ and $n_2^2$, where $n_1$ and $n_2$ are refractive indexes of core and cladding. Considering that in optical fibers the refractive indices of core and cladding are very close, one effectively has that $(n_1 - n_2)/n_1 \ll 1$. As a result, any harmonic factors $N_k$ of the function $n(\varphi)^2$ are negligible in comparison to its steady component $n^2$ and can be omitted.

As an example, the harmonics become maximal for equal alternation steps. In this case, the first harmonic, that has the maximum value of all harmonics, is equal to $A_1 = \frac{2\left(n_1^2 - n_2^2\right)}{\pi}$, while the steady component $n^2$ equals $(n_1^2 + n_2^2)/2$ and we may make an approximation as $A_1/n^2 \simeq (4/\pi)(n_1 - n_2)/n_1 \ll 1$.

Thus, for optical fibers we can always assume that $n(l)^2 \simeq n^2$. Then the system (18) will become equivalent to the following

$$
\begin{cases}
\frac{jl}{r}\overline{H_z} - j\beta\overline{H_\varphi} = jn^2\overline{E_r} \\[2mm]
j\beta\overline{H_r} - \frac{\partial \overline{H_z}}{\partial r} = jn^2\overline{E_\varphi} \\[2mm]
\frac{1}{r}\frac{\partial\left(r\overline{H_\varphi}\right)}{\partial r} - \frac{jl}{r}\overline{H_r} = jn^2\overline{E_z}
\end{cases}
\tag{19}
$$

We can then follow the analysis that we did with the conventional fibers, where $n^2$ is the average value of the $\eta^2(\varphi)$ of each layer along $\varphi$ in the $[0, 2\pi]$ interval.

### 3.2. Application to Elliptic Core Fibers

The method was applied in the calculation of fundamental modes of a fiber of elliptic core of $a$ and $b$ major and minor semi-axis, respectively, with refractive index $n_1 = 1.54$, and a cladding value of $n_2 = 1.47$ (Figure 5) for various wavelengths (defined by various V factor values $V = 2\pi b\sqrt{n_1^2 - n_2^2}$) and four ellipticity ratios $a/b$ = 1.1, 1.3, 1.5, and 2.0. Results are presented in tabulated format (Tables 1–4) compared with previous results calculated with Mathieu functions together with differences and relative differences showing a deviation which goes as only 0.01/0.123% on the average. Note the results in Tables 1–4 are not normalised the same way as in Figure 4, they are simply the odd modes, $b_{11} = \beta_o/k_o$. We can see that for small ellipticities especially, the results compare very well for all V values quite well with the Mathieu Functions results. As the ellipticity becomes very large as in Table 4, the results begin to differ. However, the accuracy of the Mathieu Functions used in this case is not known precisely, so the trend is correct and the actual difference could be debated.

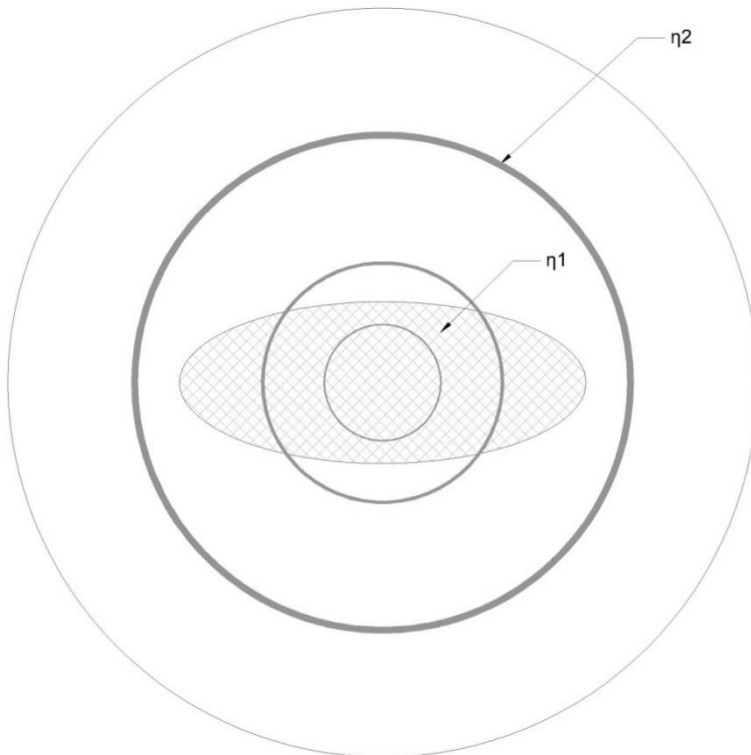

**Figure 5.** Elliptic fiber with three indicative elliptic thin layers. Inside the ellipse $r < b$ ($n = n_1$), outside the ellipse $r > a$ ($n = n_2$) and partly outside $b < r < a$ ($n_1 > n > n_2$).

**Table 1.** Comparison of ellipse for the fundamental modes.

| a/b = 1.1 | Mathieu | RTL | Differences | Relative Diff. (0/00) |
|---|---|---|---|---|
| V | $b_{11, No}$ | $b_{11, No}$ | | |
| 1.5 | 1.487454917000000 | 1.48753837672558 | −0.00008345972558 | 0.056109079 |
| 1.7 | 1.493245000000000 | 1.49345098615188 | −0.00020598615188 | 0.137945315 |
| 1.9 | 1.498457700000000 | 1.49872279558264 | −0.00026509558264 | 0.17691229 |
| 2.1 | 1.503029750000000 | 1.50331246585078 | −0.00028271585078 | 0.188097309 |
| 2.3 | 1.506994250000000 | 1.50727053366314 | −0.00027628366314 | 0.183334252 |
| 2.5 | 1.510418500000000 | 1.51067588097959 | −0.00025738097959 | 0.170403752 |
| 2.7 | 1.513376830000000 | 1.51360949624267 | −0.00023266624267 | 0.153739794 |
| 2.9 | 1.515936970000000 | 1.51614477492828 | −0.00020780492828 | 0.13708019 |
| 3.1 | 1.518160870000000 | 1.51834484072766 | −0.00018397072766 | 0.121179996 |
| 3.3 | 1.520101500000000 | 1.52026268667918 | −0.000161186679180 | 0.106036787 |

**Table 2.** Comparison of ellipse for the fundamental modes.

| a/b = 1.3 | Mathieu | RTL | Differences | Relative Diff. (0/00) |
|---|---|---|---|---|
| V | $b_{11, No}$ | $b_{11, No}$ | | |
| 1.5 | 1.491188512000000 | 1.491027765079550 | 0.000160746920450 | 0.107797853 |
| 1.7 | 1.497028990000000 | 1.496897637511800 | 0.000131352488200 | 0.087742114 |
| 1.9 | 1.502119714000000 | 1.501986592174880 | 0.000133121825120 | 0.088622647 |
| 2.1 | 1.506471523000000 | 1.506335918376860 | 0.000135604623140 | 0.090014727 |
| 2.3 | 1.510205927000000 | 1.510039049600890 | 0.000166877399110 | 0.110499764 |
| 2.5 | 1.513423500000000 | 1.513196002523000 | 0.000227497477000 | 0.150319773 |
| 2.7 | 1.516170122000000 | 1.515897403413190 | 0.000272718586810 | 0.179873342 |
| 2.9 | 1.518513480000000 | 1.518220317018870 | 0.000293162981130 | 0.193059189 |
| 3.1 | 1.520539300000000 | 1.520228501721160 | 0.000310798278840 | 0.204400030 |
| 3.3 | 1.522298190000000 | 1.521974104375010 | 0.000324085624990 | 0.212892341 |

**Table 3.** Comparison of ellipse for the fundamental modes.

| a/b = 1.5 | Mathieu | RTL | Differences | Relative Diff. (0/00) |
|---|---|---|---|---|
| V | $b_{11, No}$ | $b_{11, No}$ | | |
| 1.5 | 1.494250610000000 | 1.493636836360350 | 0.000613773639650 | 0.410756827 |
| 1.7 | 1.499922346000000 | 1.499340905733040 | 0.000581440266960 | 0.387646913 |
| 1.9 | 1.504818670000000 | 1.504203108321470 | 0.000615561678530 | 0.409060368 |
| 2.1 | 1.509039170000000 | 1.508315227121150 | 0.000723942878850 | 0.479737633 |
| 2.3 | 1.512533150000000 | 1.511793195339390 | 0.000739954660610 | 0.489215500 |
| 2.5 | 1.515493100000000 | 1.514745797603970 | 0.000747302396030 | 0.493108412 |
| 2.7 | 1.518011570000000 | 1.517265908951310 | 0.000745661048690 | 0.491209068 |
| 2.9 | 1.520166130000000 | 1.519429880525980 | 0.000736249474020 | 0.48432172 |
| 3.1 | 1.522019120000000 | 1.521299532774000 | 0.000719587226000 | 0.472784617 |
| 3.3 | 1.523620800000000 | 1.522924688766490 | 0.000696111233510 | 0.456879582 |

**Table 4.** Comparison of ellipse for the fundamental modes.

| a/b = 2 | Mathieu | RTL | Differences | Relative Diff. (0/00) |
|---|---|---|---|---|
| V | $b_{11, No}$ | $b_{11, No}$ | | |
| 1.5 | 1.499390500000000 | 1.497675992184210 | 0.001714507815790 | 1.143469840 |
| 1.7 | 1.504727250000000 | 1.502885066432660 | 0.001842183567340 | 1.224264110 |
| 1.9 | 1.509110880000000 | 1.507251064452310 | 0.001859815547690 | 1.232391584 |
| 2.1 | 1.512712190000000 | 1.510915134944720 | 0.001797055055280 | 1.187968913 |
| 2.3 | 1.515667864000000 | 1.514006539095880 | 0.001661324904120 | 1.096100896 |
| 2.5 | 1.518085750000000 | 1.516632675098700 | 0.001453074901300 | 0.957175773 |
| 2.7 | 1.520047200000000 | 1.518879733445780 | 0.001167466554220 | 0.768046252 |
| 2.9 | 1.521631209000000 | 1.520816115931370 | 0.000815093068630 | 0.535670578 |
| 3.1 | 1.522869900000000 | 1.522496060692140 | 0.000373839307860 | 0.245483418 |
| 3.3 | 1.523799100000000 | 1.523962746538420 | −0.000163646538420 | 0.107393775 |

The steady component of the refractive index for the calculations for each radius *r* is defined as $n_1$ for $r < b$, $n_2$ for $r > a$, and as $(n_1\varphi_1 + n_2\varphi_2)/\pi$ when $b < r < b$, where $\varphi_1$, $\varphi_2$ are the arcs of the circle of radius *r*, inside and outside the ellipse in the upper semi ellipse.

In the following Figure 6, the $\beta$ diagram of the fundamental even mode of an elliptic fiber with semi axis ratio $aa/bb = 2$, $n_1 = 1.54$ and $n_2 = 1.47$ and variable factor defined by: $V = bb \cdot k_0 \cdot \sqrt{n_1^2 - n_2^2}$) is shown for completeness in agreement with [32].

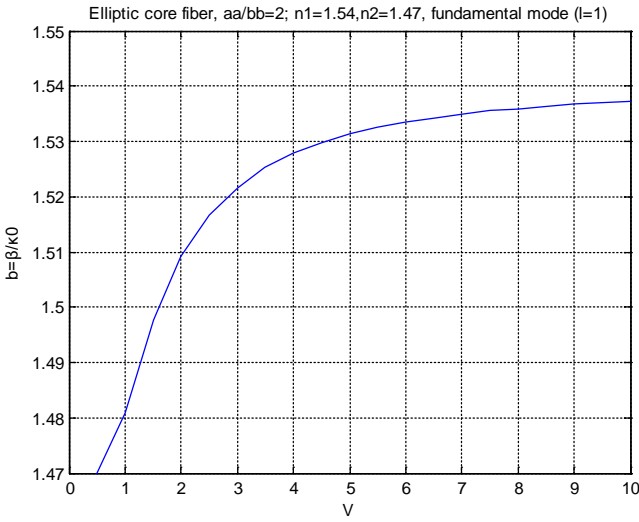

**Figure 6.** $\beta$-V diagram of the even fundamental mode an elliptic fiber of semi axis ratio $aa/bb = 2$ and core refractive index 1.54 and cladding index 1.47.

### 3.3. Application to a Rectangular Core Fiber

The method was applied also in the calculation of fundamental modes of a fiber with an rectangular core, Figure 7, with *aa* and *bb* semi-sides, with refractive index $n_1 = 1.54$, and a cladding of refractive index $n_2 = 1.47$, for various wavelengths, defined by various V factor values $V = bb \cdot k_0 \cdot \sqrt{n_1^2 - n_2^2}$ and four ratios $a/b = 1.1, 1.3, 1.5,$ and 2.0.

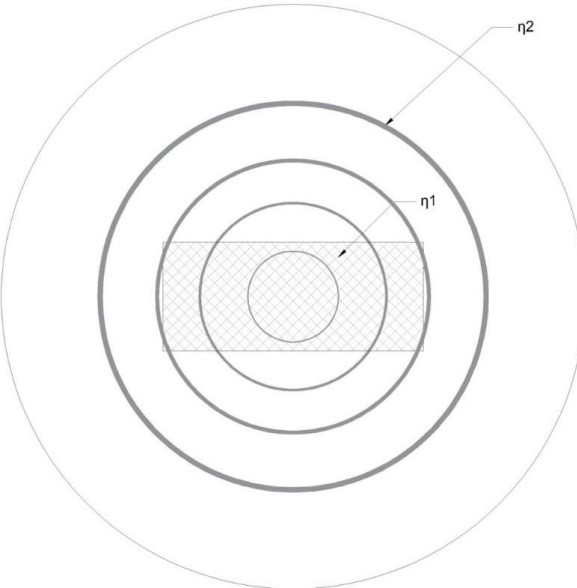

**Figure 7.** Rectangular core fiber of semi-sides *aa* and *bb*, for $r < bb$, $n = n_1$, for $r > aa$, $n = n_2$, for $bb < r < aa$, $n_2 < n < n_1$.

Birefringence results between the elliptical and rectangular waveguides are compared in tabular form in Tables 5–8 with equal semi axis. The constant component of the refractive index for calculating with each radius *r* is defined as

$$n = \begin{cases} n_1, r < b \\ \frac{1}{\pi}(n_1 \varphi_1 + n_2 \varphi_2) \\ n_2, r > b \end{cases}$$

where $\varphi_1$, $\varphi_2$ being the arcs of the circle of radius *r*, inside and outside the rectangle in the upper half plane.

**Table 5.** Comparison of birefringence for elliptical and orthogonal core fibers.

| | **Rectangular Core** | | **Elliptic Core** |
|---|---|---|---|
| **a/b = 1.1** | **Fundamental Mode Values** | **Birefringence (TR)** | **Birefringence (TR)** |
| **V** | | | |
| 1.5 | 1.492539945916010 | 0.000346704531850 | 0.000303585935370 |
| 1.7 | 1.498357447022790 | 0.000513384576600 | 0.000575417316090 |
| 1.9 | 1.503350570368440 | 0.000576764589930 | 0.000712035787010 |
| 2.1 | 1.507590076049970 | 0.000580793732230 | 0.000757892695190 |
| 2.3 | 1.511183551577450 | 0.000554099561380 | 0.000750121835720 |
| 2.5 | 1.514237341469060 | 0.000513428535340 | 0.000713597265680 |
| 2.7 | 1.516844499544290 | 0.000468033918970 | 0.000663528284040 |
| 2.9 | 1.519082616576080 | 0.000422786387590 | 0.000608733773240 |
| 3.1 | 1.521015087494710 | 0.000380083447450 | 0.000554096605350 |
| 3.3 | 1.522693317372840 | 0.000340959781570 | 0.000502161305990 |

**Table 6.** Comparison of birefringence for elliptical and orthogonal core fibers.

| | Rectangular Core | | Elliptic Core |
|---|---|---|---|
| **a/b = 1.3** | **Fundamental Mode Values** | **Birefringence (TR)** | **Birefringence (TR)** |
| **V** | | | |
| 1.5 | 1.495847758922450 | 0.000344065006270 | 0.000325722889410 |
| 1.7 | 1.501459058153440 | 0.000451639480470 | 0.000520210503450 |
| 1.9 | 1.506179273220210 | 0.000480411129310 | 0.000601842058430 |
| 2.1 | 1.510134432440230 | 0.000468166987430 | 0.000615857816190 |
| 2.3 | 1.513456882268610 | 0.000436878544030 | 0.000593498171210 |
| 2.5 | 1.516262818410440 | 0.000398422704380 | 0.000553682765940 |
| 2.7 | 1.518648025192190 | 0.000358904227450 | 0.000507155757960 |
| 2.9 | 1.520689537973830 | 0.000321272665480 | 0.000459732120240 |
| 3.1 | 1.522448794667530 | 0.000286796641510 | 0.000414381305640 |
| 3.3 | 1.523974761485380 | 0.000255871394240 | 0.000372472456950 |

**Table 7.** Comparison of birefringence for elliptical and orthogonal core fibers.

| | Rectangular Core | | Elliptic Core |
|---|---|---|---|
| **a/b = 1.5** | **Fundamental Mode Values** | **Birefringence** | **Birefringence (TR)** |
| **V** | | | |
| 1.5 | 1.498120450723510 | 0.000278361237480 | 0.000295449461120 |
| 1.7 | 1.503463400911670 | 0.000357573590440 | 0.000442306154260 |
| 1.9 | 1.507908503417930 | 0.000376605061620 | 0.000497613358440 |
| 2.1 | 1.511610102671980 | 0.000365648210230 | 0.0005013375052490 |
| 2.3 | 1.514709614765410 | 0.000341283334480 | 0.000478771792790 |
| 2.5 | 1.517324041962060 | 0.000312143747180 | 0.000444266137270 |
| 2.7 | 1.519546684126380 | 0.000282536889750 | 0.000405759261530 |
| 2.9 | 1.521451022056960 | 0.000254483058070 | 0.000367378187930 |
| 3.1 | 1.523094834995030 | 0.000228815613970 | 0.000331142666700 |
| 3.3 | 1.524523718254910 | 0.000205765806060 | 0.000297919827130 |

**Table 8.** Comparison of birefringence for elliptical and orthogonal core fibers.

| | Rectangular Core | | Elliptic Core |
|---|---|---|---|
| **a/b = 2** | **Fundamental Mode Values** | **Birefringence (TR)** | **Birefringence (TR)** |
| **V** | | | |
| 1.5 | 1.501154370588750 | 0.000115100455570 | 0.00017851280204 |
| 1.7 | 1.505905455821760 | 0.000185956233950 | 0.00028097261437 |
| 1.9 | 1.509839664490200 | 0.000217496051150 | 0.00032331073887 |
| 2.1 | 1.513123501262850 | 0.000227441962700 | 0.00033221109868 |
| 2.3 | 1.515890438928040 | 0.000225620967080 | 0.00033221109868 |
| 2.5 | 1.518243818452060 | 0.000217498908240 | 0.00030589412246 |
| 2.7 | 1.520263082984900 | 0.000206163759050 | 0.00028459934024 |
| 2.9 | 1.522009559228890 | 0.000193389032200 | 0.00026227629912 |
| 3.1 | 1.523531014266760 | 0.000180203858540 | 0.00024038749997 |
| 3.3 | 1.524865056067340 | 0.000167204354840 | 0.00021967555642 |

For the results in the Tables 5–8 we have defined birefringence as simply the difference in the orthogonal fundamental modes normalized to $k_0$, for the rectangular and elliptical waveguides. We can see the elliptical waveguide has greater birefringence in all cases of ellipticity and for all wavelengths (represented by the V value).

To make a direct comparison, the average refractive indexes as functions of *r*, for an elliptic core fiber and for a rectangular core fiber of the same aa and bb and *aa/bb* = 2, are shown in Figure 8. In Appendix B, the MATLAB codes for the average refractive indexes of an elliptic and a rectangular core fiber are given. In Figure 8 we observe that the average refractive index of the elliptical waveguide is dropping off faster than the rectangular and this facilitates the explanation of the stronger birefringence for this case.

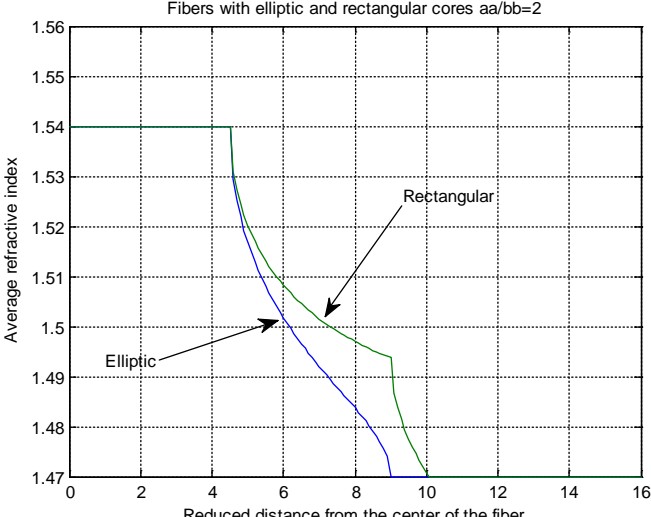

**Figure 8.** Average refractive indexes of circular thin layers of elliptic and rectangular core fibers.

### 3.4. The PCF Case

In the case of a holey fiber, we separate the whole fiber circular cross-section into a set of thin cylindrical layers, variable $\eta$ along $\varphi$ extending beyond the cladding to take into account the surrounding air with $\eta = 1$. Each layer's thickness is $\delta r = r_1 - r_2$. We can then approximate $n(r,\varphi)$ as $n(\varphi)$ for the average r $<r> = r + \delta r/2$. The refractive index can be written as a Fourier series, i.e., as $n(\varphi)^2 = \langle n \rangle^2 + \sum_{-\infty}^{+\infty} N_k \exp(jl\varphi)$. Taking into account the properties of the Fourier Transform we see that $FT(\exp(jl\phi) \cdot f(\varphi)) = f(l + l')$ so that the expressions in the second terms of Equation (18) spread around a spectrum of harmonics. This is also to be understood as a result of successive scatterings from the bored air holes. We can now use the natural geometry of the usual hexagonal lattice to see that for each set of holes we can have either $6k$ harmonics. For the fundamental harmonic of $l' = 1$, the derived harmonics passing through a layer of $6k$ holes should be $6k + 1$. Thus, for the fundamental wave crossing the successive layers it "sees" a different set of periodic rectangle functions that will be shown rigorously to contribute a different number of harmonics (7, 13, 19, . . . ).

For a common harmonic to pass through, one must then take an integer product which leads to higher and higher harmonics, thus cutting out the entire spectrum apart from the last highest frequency. We conclude that for holey optical fibers, the approximation for any of its cylindrical thin layers, $\eta(r,l)^2 \approx \eta^2(r) = \eta^2$ suffices for further analysis of the resulting equations. Thus, the original system (18) becomes

$$\begin{cases} \frac{jl}{r}\overline{H_z} - j\beta\overline{H_\varphi} = jn^2\overline{E_r} \\ j\beta\overline{H_r} - \frac{\partial \overline{H_z}}{\partial r} = jn^2\overline{E_\varphi} \\ \frac{1}{r}\frac{\partial(r\overline{H_\varphi})}{\partial r} - \frac{jl}{r}\overline{H_r} = jn^2\overline{E_z} \end{cases} \tag{20}$$

For the usual hexagonal pattern of holes, we may utilize elementary analytical geometry to derive the two separate regions where the refractive index alternates between the air refractive index $n = 1$ value and the higher value of the crystal material. We assume that along each separate layer a large circle corresponding to each cylindrical shell of radius $r$ from the center of the fiber to the center of a smaller hole of radius $r << r_0$ is cut while moving clockwise along the large circle.

Prescribing a set of circles of successive radii $r$ for each of which we can find the air holes (in 1/6 angle of the PCF) which are cut by the particular radius each time. Each arc is computed inside its respective air hole and the total sum of them divided by $\pi/3$ expresses the average squared refractive index. As a matter of fact, the square of the refractive index in this sum is equal to one, while the refractive index in the rest arc is the square of the silica refractive index. Hence the average refractive index can be easily calculated along $r$. In Figures 9 and 10, we show the average refractive index and the electric field

of a hexagonal PCF, of $n = 1.46$ with a twelve layers lattice, as functions of the reduced distance from the center of the fiber for the fundamental mode. Figure 8 was generated by a MATLAB code for air-hole diameter equal to 0.8 of the air-hole distance and the air hole diameter was 3.14 times the transmitted wave length (V = 1.607847). We also notice the parametrization used as $= (\Lambda - d/2) \times 2 \times \pi \sqrt{n_1^2 - n_2^2}$), $n_1$ for the silica refractive index, $n_2 \simeq$ minimum refractive index = 1.123, $\Lambda$ for the reduced air hole distance, d for the reduced air hole diameter, and $\Lambda - d/2$ = reduced inner core of PCF.

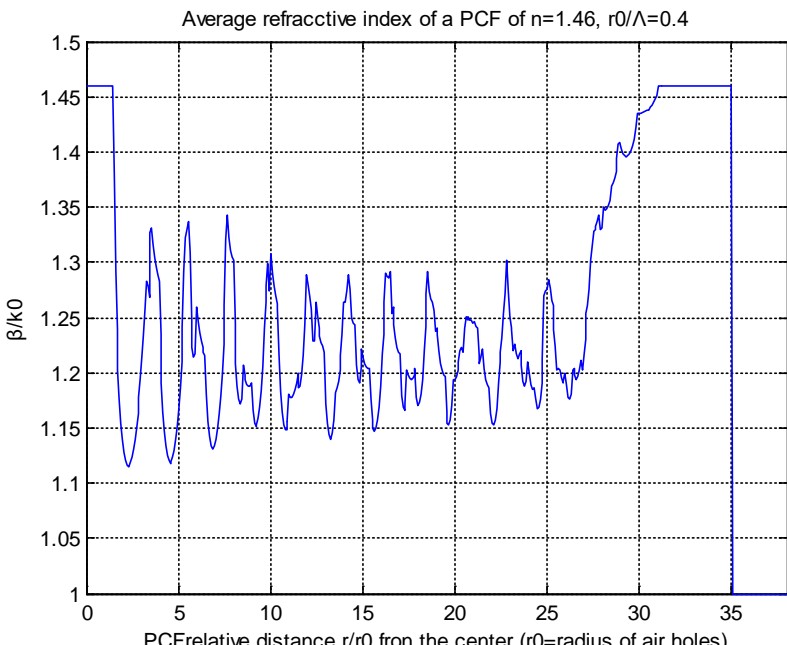

**Figure 9.** Fundamental mode $\beta/k_0 = 1.22462$, $k_0 = 1/r_0$.

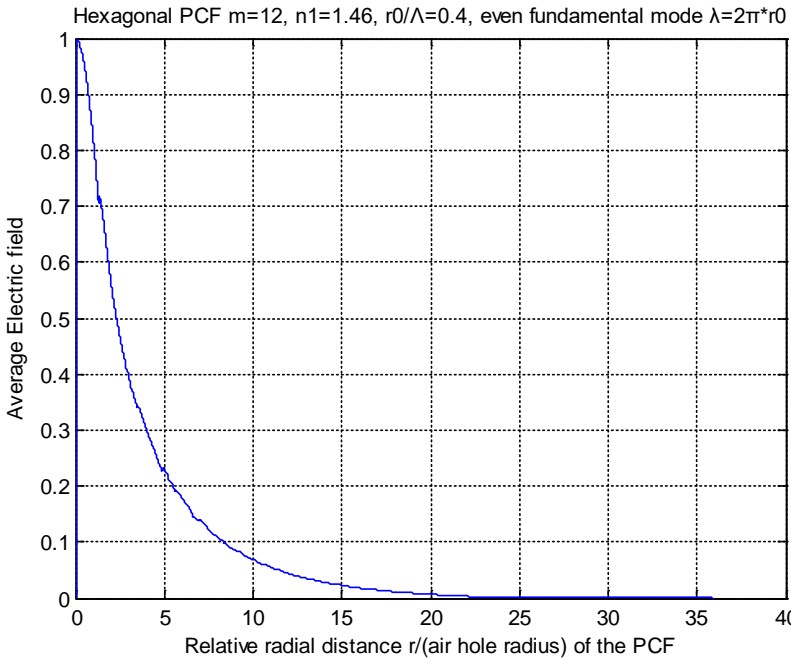

**Figure 10.** Average electric field of the even fundamental mode along the radial distance of the PCF.

For completeness we also show Figure 11, which is the $\beta/V$ diagram of the even fundamental for a PCF. This shows how easily we can produce some very useful results for a series of unconventional fibers using the transmission line method we have developed.

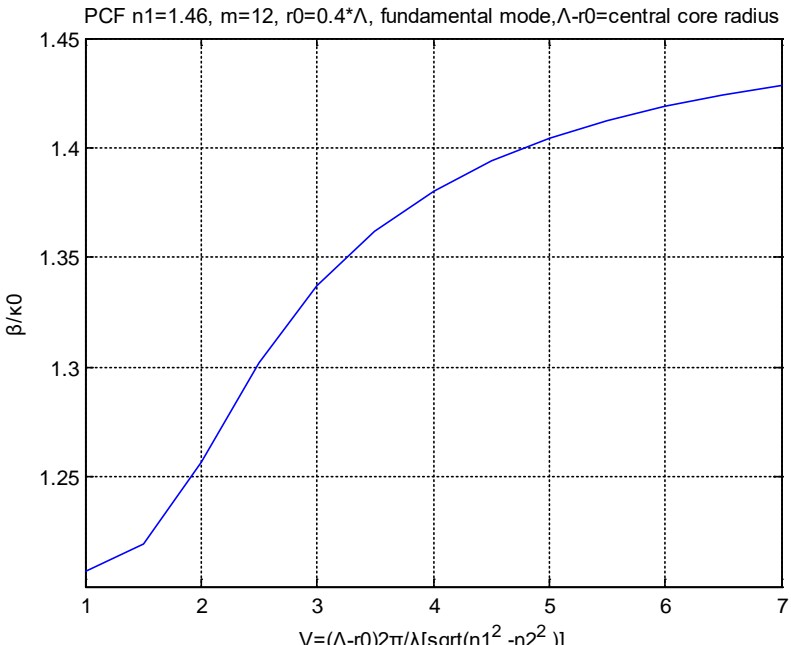

**Figure 11.** $\beta$/V diagram of the even fundamental for a PCF.

In Appendix B, the MATLAB code for the average refractive index of a hexagonal photonic crystal fiber is also given.

## 4. Conclusions

The presented resonance technique can be used for the study of unconventional fibers, i.e., fibers with cores of any shape, as long as the difference between core and cladding refractive indices is sufficiently small which holds true for almost all the monomode and holey fibers. The unconventional case is proven reducible to the same technique of conventional fibers, where for each mode order *l* we can approximate by a set of two, independent and non-homogeneous, resonant transmission lines (RTLs), each one representing one mode of the birefringence.

The simulation of unconventional fibers with RTLs gives a new, simple, and effective method for computing the eigenvalues of the RTLs representing the various modes of the holey fibers. Furthermore, for each eigenvalue, the average values of E.M. fields for every thin cylindrical layer of radius *r* of the unconventional fiber is directly computable from the relevant eigenfunctions of the RTLs.

**Funding:** This research received no external funding.

**Conflicts of Interest:** The authors declare no conflict of interest.

## Appendix A

```
function f = stepindexfun(b)
% function f = Zleft(b) + Zright(b)for step index fiber
% n1 n2 the refractive indexes of core and cladding
% tm = 0 for even modes (TM), tm = 2 for odd modes (TE), tm = 1 for average equivalent modes
% l wave number
% V factor of the fiber
% r0 = core radius × wave number
global n1 n2 tm l V
r0 = V/sqrt(n1^2 − n2^2);
```

```
N = 200;
qq = 20; % ratio of outer radius to core radius
qq0 = 200; %ratio of core radius to minimum core radius
wq = qq^(1/N);
wq0 = qq0^(1/N);
w(1) = 2*(wq0 − 1)/(wq0 + 1);
w(2) = 2*(wq − 1)/(wq + 1);
rn0 = n1;
zs(1) = −j/(rn0^tm*(abs(l) + 10^−10));
zs(2) = 0;
for n = 1:N
jj = N + 1 − n;
r1(n,1) = r0*(1/wq0^jj + 1/wq0^(jj − 1))/2;
r1(n,2) = r0*(wq^jj + wq^(jj − 1))/2;
end
j1 = 1;
for n = 1:N
r = r1(n,j1);
rn = n1;
dr = w(j1)*r;
aa1 = b^2 + (l/r)^2;
F = aa1*r;
cs = aa1 − rn^2 − 2*rn*b*l/(aa1*r^2);
zp = 1/dr/j/F/rn^tm;
zb = cs*dr/2/j/F/rn^tm;
zs(j1) = (zs(j1) + zb)*zp/(zs(j1) + zb + zp) + zb;
end
j1 = 2;
for n = 1:N
r = r1(n,j1);
rn = n2;
dr = w(j1)*r;
aa1 = b^2 + (l/r)^2;
F = aa1*r;
cs = aa1 − rn^2 − 2*rn*b*l/(aa1*r^2);
zp = 1/dr/j/F/rn^tm;
zb = cs*dr/2/j/F/rn^tm;
zs(j1) = (zs(j1) + zb)*zp/(zs(j1) + zb + zp) + zb;
end
f = imag(zs(1) + zs(2));
```

## Appendix B

```
Average refractive index of an elliptic core optical fiber
function fn = elliptic(r)
% n1 n2 refractive indexes for core and gladding
% bb minor semi axis, aa = major semi axis
global n1 n2 bb aa
if r < = bb;
fn = n1;
elseif r > bb && r < aa;
```

```
c1 = (1/r^2 − 1/aa^2)/(1/bb^2 − 1/r^2);
cc = 2*atan(sqrt(c1));
fn = sqrt((n2^2*(pi − cc) + cc*n1^2)/pi);
else fn = n2;
end
Average refractive index of a rectangular core optical fiber
function fn = rectangular(r)
% n1 n2 refractive indexes for core and cladding
% bb minor semi axis, aa = major semi axis
global n1 n2 bb aa
cc = sqrt(aa^2 + bb^2);
if r < = bb;
fn = n1;
elseif r > bb && r < aa;
c = 2*asin(bb/r);
fn = sqrt((n2^2*(pi − c) + c*n1^2)/pi);
else fn = n2;
end
if r > = aa & r < cc; c = 2*(asin(bb/r) − acos(aa/r));
fn = sqrt((n2^2*(pi − c) + c*n1^2)/pi);
end
Average refractive index of a photonic crystal holey core optical fiber
function ref = holey(r)
%PCF hexagonal
% n1 = silica refractive index
% m = number of lattice rows of air holes
% d = reduced value of the distance of air hole centers
% ro = reduced radius of air holes r0 < 0.5*d
% R = external radius of the fiber gladding R > m*d
% If not given R = d*(m + 2), after R it is taken as the air value
global n1 m d r0 R
if r < = d − r0;
ref = n1;
elseif r > d − r0 && r < m*d + r0;
for nn = 1:m;
for n = 1:nn; rr(nn,n) = nn*d*exp(j*2*pi/3) + (n − 1)*d;
rt(nn,n) = abs(rr(nn,n));
end
end
f = 0;
for nn = 1:m;
for n = 1:nn; rrr = rt(nn,n); drt = abs(rrr − r);
if drt < r0; ff = 2*acos((r^2 + rrr^2 − r0^2)/2/r/rrr); f = ff + f; end
end
end
ref = sqrt((f + (pi/3 − f)*n1^2)/(pi/3));
else ref = n1;
end
if r > = R; ref = 1;
end
```

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
