# Peer review of "Resonant Transmission Line Method for Unconventional Fibers"

_applsci, doi:10.3390/app9020270_

Reviewer 1 Report

Review of the manuscript applsci-409774

 Resonant transmission line method for unconventional fibers

By A.    C. Boucouvalas, C.D. Papageorgiou, E. Georgantzos, T. E. Raptis

The authors in their previous works developed an efficient method of the optical fibers analysis based on the transmission line model. In such a case, the Maxwell equations for the electric and magnetic fields of the optical fiber modes are replaced by the so-called telegrapher’s equations for the voltage and current in the equivalent transmission line. The transmission line approach has been used successfully for the radio and microwave signals propagating in the rectangular and circular waveguides, microstrip lines, etc. The transmission line formalism is much simpler from the mathematical point of view as compared to the Maxwell equations because it concerns the scalar functions in the one-dimensional case instead of vector functions in the three-dimensional case.

In the proposed manuscript the authors applied their method to the unconventional optical fibers with non-circular, non-symmetric, or eccentric cores. They derived the corresponding equations, presented the MATLAB functions and codes, and carried out the numerical simulations.  The simulation results show an excellent agreement with the calculation results based on the Mathieu functions.

The proposed paper is clearly written, contains novel results and can be interesting for the researchers and engineers occupied in the optical communications.

The paper can be recommended for publication in Applied Sciences after the following minor revisions.

1.      The references include only the papers of the authors of the proposed manuscript. Please, add some references of other authors publications.

2.      Please, add to section 4 Conclusion the comparison of the results obtained with the experimental and theoretical results of other authors published in peer reviewed journals and/or monographs.

3.       Please, enumerate the tables (lines after 239 – before 243, after 261 - before 262) and write some comments for the numerical results presented in these tables.

4.      There are some misprints in the text. See for instance, page 1, line 32:. Page 1, line 33:  . Page 2, line 49: “…lattice of seven rows…”. It seems that the number of rows in figure 2 is different. Line 259: . Line 291:  . Line 302:  for the.

Author Response

We would like to  thank the reviewer for pointing out details tha make the text substantially better.

Comment 1:

We absolutely agree on your comment and we have added 13 extra references relevant to our work briefly discussed at the introduction which has been revised.

Comment 2:

For elliptical fibers we have included some works in the main body mainly based on the use of Mathieu functions. For the rest of waveguide types we unfortunately do not have access to similar results.

Comment 3: Explanatory text has been added to the relavant sections for all tables.

Comment 4: The indicated text has been corrected accordingly

Reviewer 2 Report

This work claims to present a general review of the resonant transmission line method to study conventional and unconventional optical fibers. Although the authors address detailed simulations concerning the application of the technique to different optical fiber geometries, the text itself lacks organization and is not so clear. It also doesn't provide enough information as a review paper, specially in the abstract and introduction sections. I suggest the authors to re-write these sections in order to make it less confusing and explaning clearly the structure of the manuscript,  the concept of resonant transmission line and its proposed application. Also, the simulation results should be better discussed and not just exhibited. It would be interesting as well if the authors could include more references of previous work that are not their own, if possible. 

Author Response

We thank the reviewer for helping in substantially improving the quality of the main text.

We agree that the paper is not a full review of everything in existing literature. If this was the case it would take many pages to summarize other methods. We are only revisit our methods and reformulate the Transmission Line problem it in a manner that allows fully separating the coupled differential equations and show clearly even the birefringence of a circular core fiber. This result alone is significant for cases where polarization mode dispersion is of interest in high data rate communication systems and for sensing applications where special shape fibers are used. We have tried to explain and make the paper structure better. We made our best to improve readability ans also, added further explanatory text on all simulated results. Also, new references were added to make trasnparent the origin and evolution of parts of the methodology used.

Round  2

Reviewer 2 Report

The authors improved the manuscript significantly, thus I have no more suggestions.